# Application of an Integrated System of Thermal Pressure Hydrolysis/Membrane Techniques to Recover Chromium from Tannery Waste for Reuse in Hide Tanning Processes

**DOI:** 10.3390/membranes13010018

**Published:** 2022-12-23

**Authors:** Anna Kowalik-Klimczak, Maciej Życki, Monika Łożyńska, Christian Schadewell, Thomas Fiehn, Bogusław Woźniak, Monika Flisek

**Affiliations:** 1Łukasiewicz Research Network–Institute for Sustainable Technology, Pułaskiego St. 6/10, 26-600 Radom, Poland; 2Prüf- und Forschungsinstitut Pirmasens e.V., Marie-Curie-Str. 19, 66953 Pirmasens, Germany; 3Countrywide Chamber of Leather Industry, Włodzimierza Krukowskiego St. 1, 26-600 Radom, Poland

**Keywords:** microfiltration (MF), nanofiltration (NF), thermal pressure hydrolysis (TPH), chromium(III), tannery waste, circular economy

## Abstract

This paper presents the results of research on a new method of chromium recovery from solid waste generated during the tanning of raw hides. In the first stage, the shredded mixture of useless leather scraps is decomposed through thermal pressure hydrolysis (TPH) in nitric acid in appropriate process conditions. Then, the liquid product of this process (hydrolysate) is fractionated using membrane separation techniques. The microfiltration (MF) process enables the initial purification of the hydrolysate by concentrating the organic matter. On the other hand, the nanofiltration (NF) process enables a three-fold concentration of total chromium in the pre-purified hydrolysate. The total chromium concentrate prepared in the above manner was successfully used in the model tanning processes. These processes were carried out on pickled bovine hides, using a mixture of a commercial chromium tanning agent and chromium concentrate after nanofiltration. The reference sample was bovine hide traditionally tanned with a commercial chromium tanning agent. Based on the results of the physical and chemical analyses, it was found that the properties of hides tanned using chromium recovered from waste are similar to those of hides tanned traditionally using a commercially available chromium tanning agent. The industrial implementation of the developed tannery waste valorisation technology would enable transition from a linear economy to circular economy.

## 1. Introduction

Animal hide is a troublesome waste product generated by the meat industry, which, through the use of proper tanning processes, can be transformed into a product that can be used to manufacture footwear, clothing, furniture, and car upholstery [1]. Unfortunately, leather production processes generate significant amounts of waste. It is assumed that 1 ton of raw hides produces about 200 kg of leather, while the remaining 80% or so of the initial weight is, in fact, post-production waste, mainly in the solid form. Such waste can be grouped into two categories: raw and tanned. Raw waste is generated before the tanning process and it includes raw skin trimmings (i.e., tissues separated from the hide) and raw skin cuttings (i.e., useless pieces of hide) [2]. Tanned waste is generated after the tanning process and it includes shavings, tanned hide cuttings (i.e., useless leather pieces), and dust (i.e., waste generated from various operations, e.g., polishing) [3,4]. Untanned waste disposal is not a problem, as this waste is used to produce, among other things, gelatine, adhesives, animal feed, and fertilizers [5,6].

However, safe management of tannery waste containing chromium compounds remains challenging for technologists and engineers alike. Therefore, many R&D centres in Poland and abroad are working hard to develop an effective way to manage this type of waste. One promising technology for managing tannery waste containing chromium assumes the use of a phase transfer catalyst. This technology enables the recovery of chromium(III) in the form of inorganic compounds that can be directly used in the process of hide tanning. On the other hand, by-products of these processes that have the form of residues (e.g., collagen) can be used to produce technical gelatine, organic fertilizers, or biopolymers [7]. This method, however, requires high investment costs and thus it has not yet been commercialised. An alternative approach to managing chromium waste generated by the tanning industry involves thermal inactivation enabling energy recovery and concurrent waste volume reduction. The disadvantage of this approach lies in the emission of carcinogenic chromium compounds [8]. Only the use of a tunnel furnace and the selection of proper process conditions make it possible to limit the emissions of hazardous waste gases and recover chromium for future reuse [8,9]. Interesting technological solutions in this regard also include a thermochemical method, which enables chromium waste generated by the tanning industry to be used to produce sorption materials designed to remove common pollutants present in wastewater [10,11], and an ultrasound chromium extraction from tannery wastewater [12]. Recent years have also seen a development of a tannery waste disc granulation method, the industrial application of which could reduce the costs of waste disposal and transport. However, it was only proven to be effective in the case of shavings [13]. As a result, unmanaged tannery waste still fills landfills, which means that the carcinogenic forms of chromium(VI) it contains may enter the environment [14,15,16]. Given the above, new material recycling initiatives must be taken in order to enable the development of an effective management of tannery waste containing chromium.

The article is centred around the development of a new method for the valorisation of chromium waste generated by the tanning industry. The method assumes that the mixture of chromium waste generated by tanneries should first be broken down through thermal hydrolysis in properly selected process conditions in an acidic environment [17]. Then, the liquid compound produced by hydrolysis (hydrolysate) should be treated using membrane filtration methods. Based on the literature findings [18,19,20,21,22], the authors suggest that nanofiltration should be used for the purpose of the concentration of chromium ions in liquids after the thermal hydrolysis of chromium waste generated by tanneries carried out in an acidic environment. On the other hand, taking into consideration the relationship between the degree of concentration of chromium solutions and the organic content of tannery wastewater [19], hydrolysates should be pre-treated with the use of microfiltration. This process enables the separation of the total suspended solids and components responsible for chemical oxygen demand, organic nitrogen, and fats [21]. The proposed technological solution will help recover chromium compounds to be reused to tan hides, and to concentrate the organic matter (fats and proteins) for the purpose of energy production. The research described in the article was conducted in laboratory and industrial conditions in cooperation with Polish tanneries.

## 2. Materials and Methods

### 2.1. Thermal Pressure Hydrolysis (TPH)

The thermal pressure hydrolysis (TPH) was carried out at a Eugen Schmitt GmbH 100 dm^3^ pilot plant. The mixture of chromium waste from tanneries (shavings, cuttings, and dust) was fed to the tank of TPH pilot plant together with a 1% solution of nitric acid(V). During the first few pressure cycles, the temperature inside the tank reached 117 °C, and during the second process, it increased to 160 °C and was maintained at that level for a total of 60 min, after which the tank was left to cool down. The TPH was carried out on a mixture of 2 kg of waste and 10 kg of nitric acid(V) solution. Between 5 and 10 kg of steam was used for heating and cooling.

### 2.2. Membrane Filtration

Membrane processes employed a laboratory installation to test the filtration and separation properties of flat-sheet membranes in a cross-flow SEPA CF II type system, as described in articles [23,24]. The liquid fraction from the thermal hydrolysis of the tannery waste containing chromium was placed in a feed tank and pumped to a membrane module, where it was separated into a permeate, transferred to a separate tank, and the retentate (concentrate) was redirected back to the feed tank. Each process lasted until the initial volume of the feed was reduced twice (5 dm^3^).

Micro- and nanofiltration processes employ commercial membranes, the basic parameters of which are presented in Table 1.

### 2.3. Physical and Chemical Parameters of Waste, Hydrolysates, and Membrane Filtration Products

A multiparameter analysis of the physical and chemical properties was carried out to characterise the chromium waste from the tanneries, the liquid fraction obtained as a result of the thermal hydrolysis of the tannery waste (acid hydrolysates), and the liquid streams resulting from the membrane filtration processes. The conductivity and pH of the liquid were measured with a Mettler Toledo SevenExcellence benchtop meter. The chemical oxygen demand was determined using a LAR QuickCODlab analyser. Measurements were based on high-temperature sample combustion at 1200 °C to ensure a high accuracy and reliability of readings. The total carbon and total nitrogen bound concentrations were determined using an Elementar vario TOC cube analyser. The analysis involved measurement of the carbon dioxide emitted as a result of high-temperature catalytic combustion of the sample exposed to an oxygen stream. The concentration of chlorides and sulphates was determined using LCK cuvette tests and a Hach Lange UV-VIS DR 6000 spectrophotometer. The dry matter content of the waste, hydrolysates, and membrane filtration products (permeate and retentate streams) was determined using a moisture analyser based on the difference in weight before and after drying at 105 °C. On the other hand, the organic dry matter content was determined based on the difference in weight before and after roasting at 550 °C and with the use of a Binder FD 23 drying oven, a Thermo Scientific FB1410M-33 Compact Benchtop Muffle Furnace, and an Ohaus analytical balance with a certified readability of 1 mg. The chromium(VI) concentration was determined through spectrophotometry (UV–VIS) with 1,5-diphenylcarbazide after the sample was leached using an etching solution in a mixture of sodium hydroxide and calcium carbonate. To determine the concentration of total chromium, the sample was first mineralised in a mixture of sulphuric acid and hydrogen peroxide. Thus, the obtained mineralisates were analysed using inductively coupled plasma optical emission spectroscopy (ICP-OES) with a Perkin Elmer Optima 5300 V.

### 2.4. Model Tanning Tests

After decalcification, the bovine hides were tanned in a pickling bath. Model tanning tests were conducted on an industrial scale using a commercial tanning agent (Chromitan B) and chromium recovered from waste subject to thermal pressure hydrolysis. Model tanning tests were carried out using a mixture of retentate after NF (3%) and commercial tannin agent (5%). The reference sample was traditionally tanned cowhide with a commercial chrome tannin (7%). After the tanning process, the hide samples were removed from the drums and left for 48 h. Then, their physical and chemical parameters were analysed, i.e., thickness (PN-EN ISO 20344), tensile strength (PN-EN ISO 3376), elongation (PN-EN ISO 3376), tear strength (PN-EN ISO 3377-2), bursting factor using a lastometer (PN-EN ISO 17693), adhesion of finish (PN-EN ISO 11644), and chromium(III) content converted to Cr_2_O_3_ (PN-EN ISO 5398-1).

### 2.5. Microscope Tests

As part of the microscope tests, the polymer membranes and bovine hides were analysed. Their surface was imaged using a Hitachi SU-70 Schottky Field Emission scanning electron microscope (SEM). Analyses were carried out in a vacuum (1 × 10^−8^ Pa) with an acceleration voltage of 15 kV, inclination angle of 30.4°, and a secondary electron detector. The chemical composition of the polymer membranes and bovine hides was analysed using a Thermo Scientific X-ray EDS microanalyzer. Prior to the SEM/EDS tests, a 30 nm layer of carbon was sputtered on the surface of the samples using a BAL-TEC SCD 050 Sputter Coater for coating the dry samples with a conductive material. In parallel, microscopic analyses of the surface of the bovine hide samples were carried out using a Keyence VHX 6000 digital microscope.

## 3. Results and Discussion

### 3.1. Hydrolysis of Tannery Waste Containing Chromium

First, chromium waste generated by tanneries in the form of shavings, cuttings, and dust (Figure 1) was subject to thermal pressure hydrolysis.

The physical and chemical parameters of the chromium waste from tanneries are presented in Table 2. The obtained results varied among the analysed sample batches and within the same batch as well. Based on the results of the physical and chemical parameter analysis, the authors decided that the samples characterised by a low moisture content (<20%) would not be dried before treatment, and those with a higher moisture content would be air-dried before hydrolysis.

As a result of the TPH, the tannery waste containing chromium decomposed into liquid fraction, which was then analysed for the content of dry matter, dry organic matter, and chromium. The analysis results are presented in Table 3. As a result of the TPH, dry mass and dry organic mass decomposed, leaving traces of solids and a high chromium(III) content in the solution.

### 3.2. Hydrolysate Treatment Using Micro- and Nanofiltration

The next stage of the analysis concerned the characterisation of the physical and chemical parameters of the liquid fraction resulting from the thermal pressure hydrolysis of the tannery waste containing chromium carried out in an acidic (nitric acid(V)) environment, i.e., acid hydrolysate (Table 4). This was followed by membrane treatment of the acid hydrolysate in accordance with the procedure presented in Figure 2.

The microfiltration took place under a pressure of 2.0 bar and feed flow of 180 dm^3^/h, and it employed an JX membrane characterised by an average performance of 61.6 ± 7.6 dm^3^/(m^2^h). During microfiltration, the performance dropped by 28%, which was caused by the deposition of filtered liquid components on the membrane surface (Figure 3). The EDS analysis of the mass composition (Figure 3) showed that the membrane pollutants were mainly composed of carbon, oxygen, chromium, nitrogen, and sulphur. Most probably, these pollutants were an organic matter bound to chromium, which did not completely decompose during the thermal hydrolysis carried out in an acidic (nitric acid(V)) environment.

Microfiltration enabled an initial hydrolysate treatment through the concentration of organic matter (fats and proteins) to the level that would allow it to be used for energy production purposes [25,26,27]. The microfiltered permeate was then subject to nanofiltration. This process took place under a pressure of 20 bar and feed flow of 180 dm^3^/h, and it employed TS40 and DL membranes characterised by an average performance of 42.7 ± 5.1 dm^3^/(m^2^h) and 51.8 ± 7.6 dm^3^/(m^2^h), respectively. During nanofiltration employing a TS40 membrane, the performance dropped by 27% and in the case of a DL membrane, by 33%. The membrane surface before and after nanofiltration was analysed using the SEM/EDS method (Figure 4 and Figure 5).

The SEM/EDS analyses (Figure 4 and Figure 5) showed that the components of the filtered liquid were deposited on the surface of both nanofiltration membranes, with more of them deposited on the surface of the DL membrane. As a result, the TS40 membrane was selected for further tests. Using this membrane, a two-stage nanofiltration process was carried out, which enabled a three-fold concentration of chromium in the pre-treated hydrolysate, with its concurrent retention at a level of 99% (Figure 6).

### 3.3. Model Bovine Hide Tanning Test Using Chromium Recovered from Tannery Waste

The procedure for applying chromium recovered from waste generated during hide tanning is presented in Figure 7. Chromium recovered from tannery waste (in the form of retentate from nanofiltration of acidic hydrolysate) was used to prepare a tanning bath with a total chromium concentration of 11 g/dm^3^ and pH ≈ 2.5. The model hide tanning tests were carried out in industrial conditions using a mixture of chromium concentrate and a commercial chromium tanning agent. Bovine hide tanned in a traditional manner using a commercial chromium tanning agent was used as a reference sample.

At the time of the visual inspection of samples, it was found that the colour of the semi-finished leather product tanned using a chromium retentate from the nanofiltration of acidic hydrolysate was grey and brown, which means that this material could only be dyed black or dark brown. On the other hand, the semi-finished product tanned using only a commercial tanning agent had a blue-grey colour. At the next stage of the study, both leather semi-finished products were subject to finishing. This was followed by an analysis of their physical and chemical properties, as part of which the following were verified: thickness, tensile strength, elongation, tear strength, bursting factor, adhesion of finish, and chromium(III) content. The analysis showed that the hides tanned using chromium recovered from waste and the hides tanned in a traditional manner had similar physical and chemical properties (Table 5).

Samples of hides tanned using chromium recovered from waste were analysed under a microscope, and their microscopic images were compared to the microscopic images of the hides tanned in a traditional manner using a commercial chromium tanning agent. For the purpose of microscope tests, 3D and SEM microscopes were used. The images of the hides tanned in a traditional way and using chromium concentrate recovered from waste are presented in Figure 8, and the maps of chromium distribution on the surface of the hides tanned in a traditional way and using chromium concentrate recovered from waste, created using the energy dispersive spectroscopy (EDS) technique, are presented in Figure 9. Microscopic observations showed that the use of chromium recovered from waste does not adversely affect the structure of the hide surface (Figure 8) and that chromium was distributed on the surface of the hides tanned using chromium recovered from waste evenly and similarly, as in the case of hides tanned in a traditional way (Figure 9).

Bovine hide tanned using chromium recovered from the tannery waste with the proposed technology was used to make a prototype of men’s footwear for everyday use (Figure 10).

However, the prototype must be tested for its hygienic and performance properties, such as water permeability, weather resistance, or health safety [28,29]. Positive results from such tests would make it possible to obtain the necessary approvals required to implement the technological solution developed by the authors and discussed in this paper into industrial production. The evaluation of the hygienic and performance properties of the prototype footwear is the topic around which the authors’ future cooperation will revolve.

## 4. Conclusions

The developed concept of the technology for the valorisation of tannery waste containing chromium assumes the use of an integrated thermal pressure hydrolysis system and membrane techniques to recover chromium compounds. First, the mixture of useless scraps (cuttings, shavings, and dust) of hides tanned using chromium is broken down through thermal pressure hydrolysis in properly selected process conditions in an acidic environment (nitric acid(V)). Then, the liquid compound produced by hydrolysis (hydrolysate) is fractionated using membrane filtration methods. Microfiltration enabled the initial hydrolysate treatment through the concentration of the organic matter (fats and proteins). On the other hand, two-stage nanofiltration enabled total chromium retention at a level of 99% and a three-fold concentration of total chromium in a pre-treated hydrolysate. Chromium recovered from waste in the manner described in the article was successfully used in tanning processes. The leather thus obtained was compared with leather tanned with a commercial tanning agent. This study has shown that leathers tanned with waste chrome and traditionally tanned leathers do not differ significantly in terms of their physical and chemical properties. In order to confirm the usefulness of leather tanned with a tanning agent obtained from the proposed technology of chromium recovery, a prototype of men’s footwear was made.

## Figures and Tables

**Figure 1 membranes-13-00018-f001:**
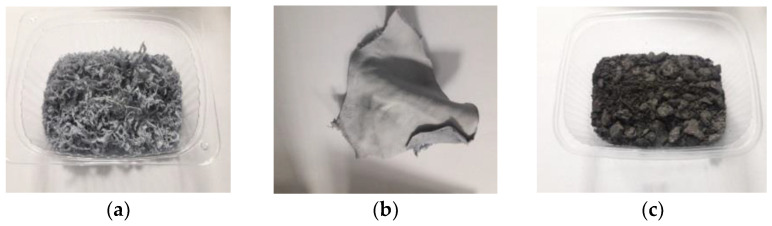
Images of the samples of the chromium tannery waste in the form of (**a**) shavings, (**b**) cuttings, and (**c**) dust.

**Figure 2 membranes-13-00018-f002:**
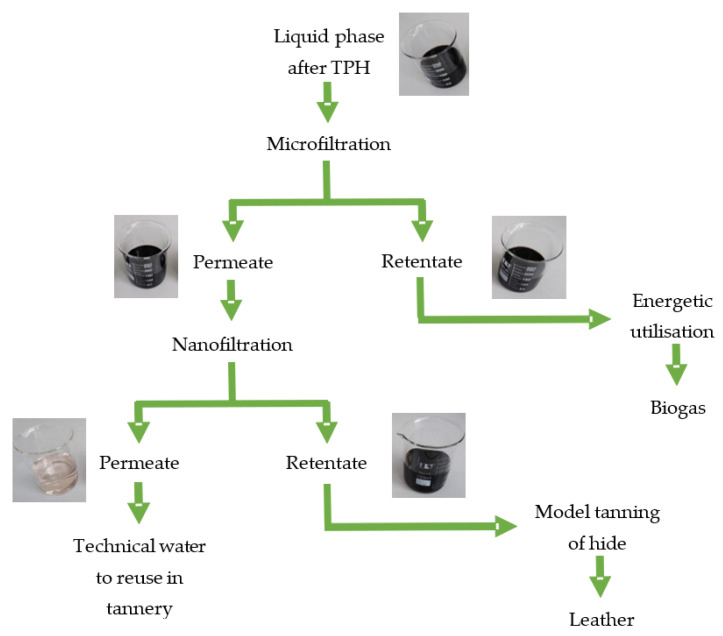
Fractionation procedure for a liquid fraction created as a result of the thermal pressure hydrolysis of chromium waste from tanneries carried out in an acidic environment.

**Figure 3 membranes-13-00018-f003:**
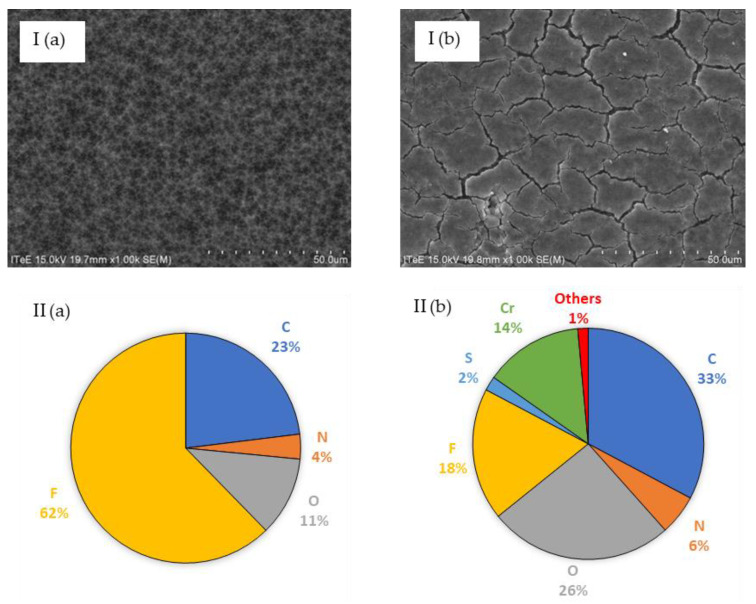
Images of SEM (I) (magn. ×1000) and elemental composition of analysis EDS (II) of the surface JX membrane before (**a**) and after (**b**) microfiltration of the liquid fraction created as a result of the thermal pressure hydrolysis of chromium waste from tanneries.

**Figure 4 membranes-13-00018-f004:**
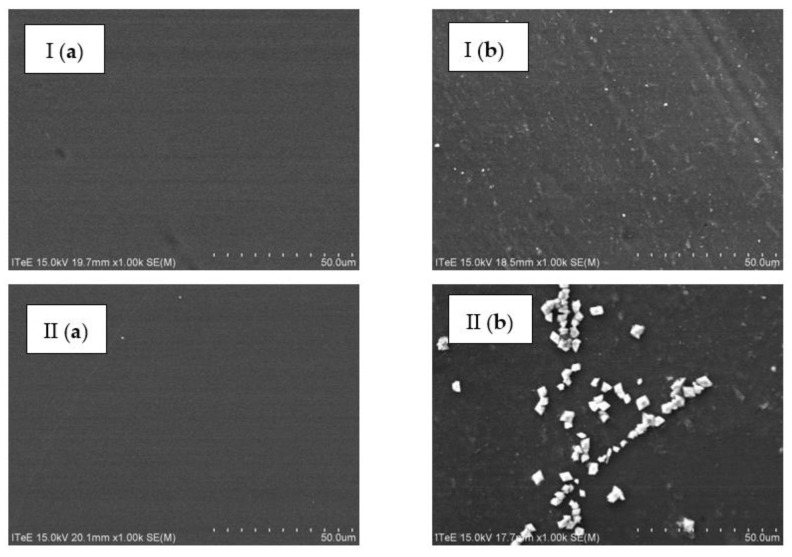
Images of the surface of the (I) TS40 and (II) DL membranes before (**a**) and after (**b**) nanofiltration of the liquid fraction created as a result of the thermal pressure hydrolysis of chromium waste from tanneries, pre-treated using microfiltration, taken with SEM (magnification: ×1000).

**Figure 5 membranes-13-00018-f005:**
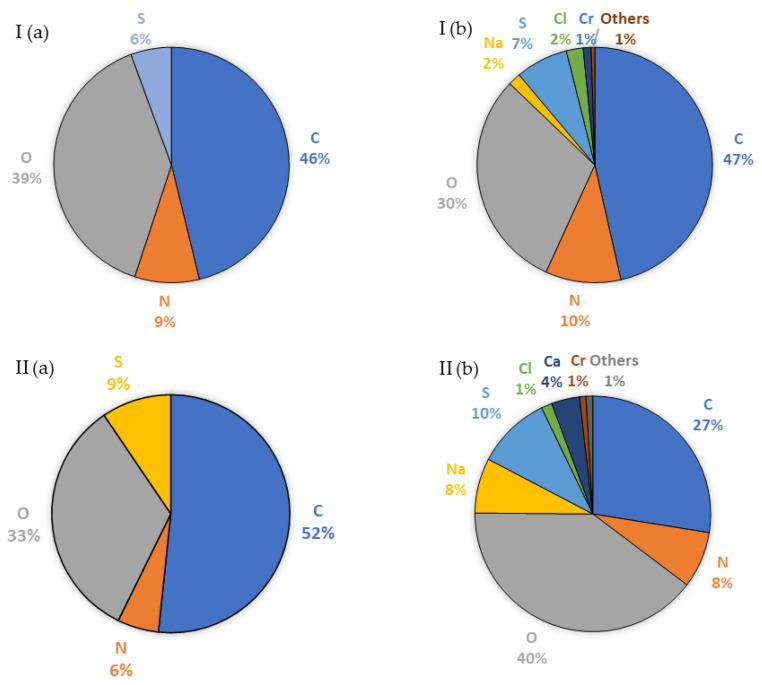
Elemental composition (%) of the surface of (I) TS40 and (II) DL membranes before (**a**) and after (**b**) nanofiltration of the liquid fraction created as a result of the thermal pressure hydrolysis of chromium waste from tanneries, pre-treated using microfiltration, determined using EDS.

**Figure 6 membranes-13-00018-f006:**
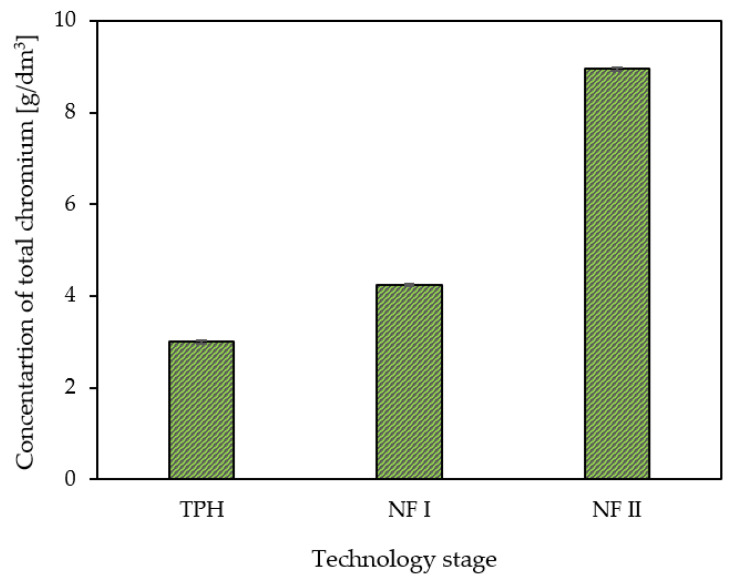
Total chromium concentration after individual stages of the proposed tannery waste valorisation technology. TPH—thermal pressure hydrolysis; NF I—nanofiltration (stage I); NF II—nanofiltration (stage II).

**Figure 7 membranes-13-00018-f007:**
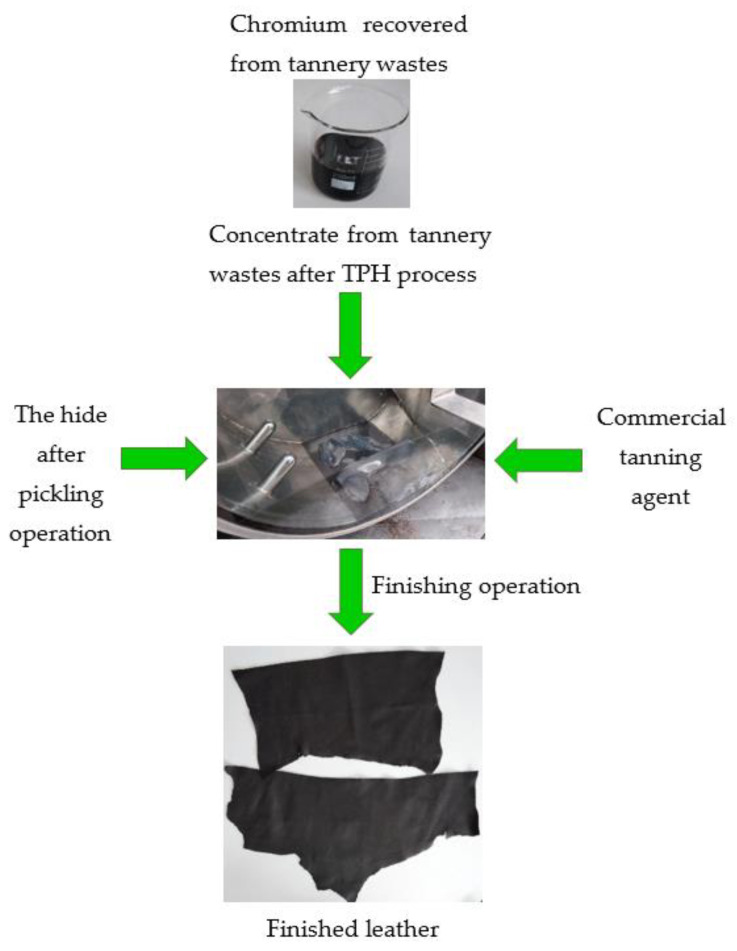
Procedure for using chromium recovered from waste generated during hide tanning.

**Figure 8 membranes-13-00018-f008:**
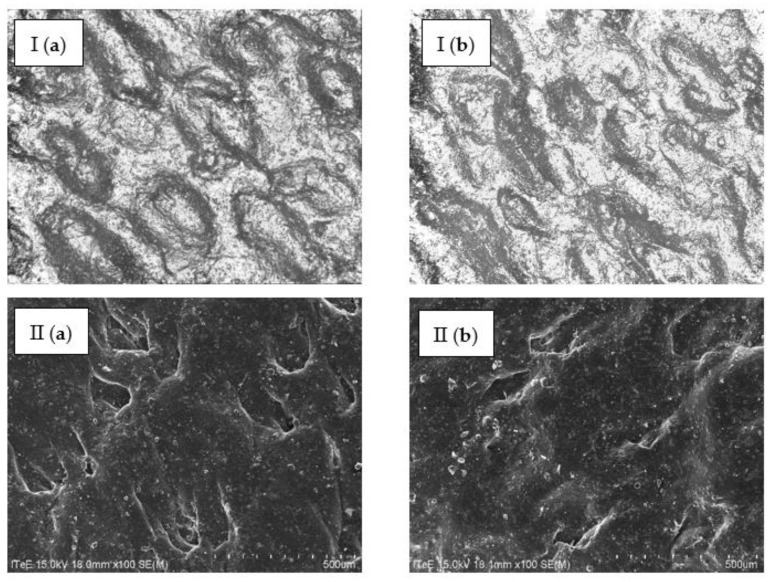
Images of the surface of bovine hides tanned using (**a**) a commercial tanning agent and (**b**) chromium recovered from tannery waste (concentrate), taken using the (I) 3D microscope (magnification: ×500) and (II) SEM (magnification: ×100).

**Figure 9 membranes-13-00018-f009:**
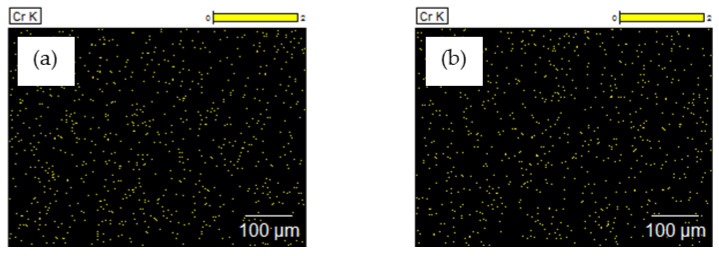
Maps of chromium distribution on the surface of the bovine hides tanned using (**a**) a commercial tanning agent and (**b**) chromium recovered from tannery waste (concentrate), taken using EDS.

**Figure 10 membranes-13-00018-f010:**
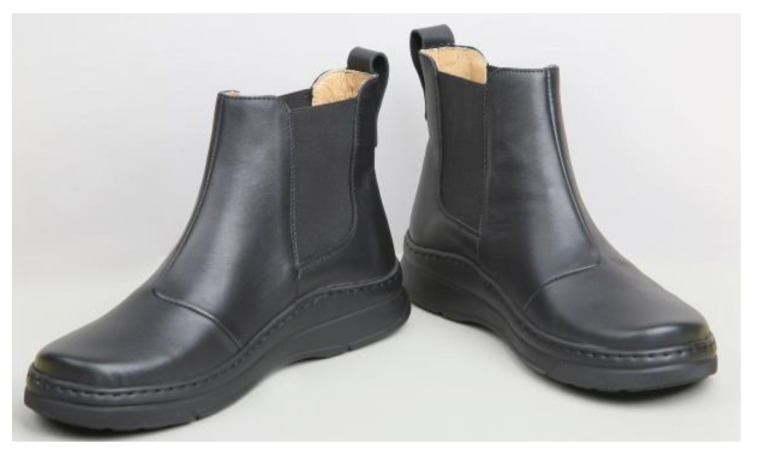
Prototype of men’s footwear for everyday use made of bovine hide tanned using chromium recovered from tannery waste.

**Table 1 membranes-13-00018-t001:** Parameters of membranes used to treat liquid obtained as a result of the hydrolysis of tannery waste containing chromium carried out in an acidic environment.

Type of Membrane	MF JX	NF TS40	NF DL
Manufacturer	SUEZ(GE)	TriSEP	SUEZ(GE)
Material	PVDF	PPZ	PA
Pore size [µm]	0.3	-	-
Cut-off [Da]	-	~200	~150–300
pH range	1–11	2–11	2–10
Max. temperature [°C]	45	45	45

PVDF—polyvinylidene fluoride; PPZ—poly(piperazine-amide); PA—polyamide.

**Table 2 membranes-13-00018-t002:** Characteristics of tannery waste containing chromium.

Parameter	Shavings	Cuttings	Dust
Dry matter content [%]	79.8–80.7	83.6–87.3	90.9–91.6
Organic dry matter content [% dm]	83.9–87.7	82.2–93.5	86.1–86.5
Total nitrogen bound [% dm]	13.0–15.1	13.0–14.5	7.7–8.0
Total chromium [% dm]	2.74–3.30	2.88–3.86	2.25–23.38

**Table 3 membranes-13-00018-t003:** Results of the physical and chemical parameter analyses of the liquid phase after thermal pressure hydrolysis of the tannery waste containing chromium.

Tanning Waste Type	Recovery [% of feed]		
	Dry Matter	Organic Dry Matter	Total Chromium
Cuttings	89	99	85
Shavings	95	101	81
Dust	33	34	9

**Table 4 membranes-13-00018-t004:** Physical and chemical parameters of the liquid fraction created as a result of the thermal pressure hydrolysis of chromium waste from tanneries carried out in an acidic environment.

Parameter	Value
pH	2.515 ± 0.004
Conductivity [mS/cm]	26.89 ± 0.02
Total chromium [g/dm^3^]	3.00 ± 0.01
Chromium (VI) [mg/dm^3^]	24.20 ± 0.35
Chemical oxygen demand [g O_2_/dm^3^]	115.4 ± 5.7
Total organic carbon [g/dm^3^]	29.11 ± 0.27
Total nitrogen bound [g/dm^3^]	19.31 ± 0.23
Chlorides [g/dm^3^]	4.79 ± 0.16
Sulphates [g/dm^3^]	7.447 ± 0.042
Dry matter content [%]	10.15 ± 0.01
Organic dry matter content [% dry matter]	85.71 ± 0.21

**Table 5 membranes-13-00018-t005:** Comparison of the physical and chemical parameters of hides tanned in a traditional manner and using chromium recovered from waste subject to acid hydrolysis.

Parameter	Sample 1	Sample 2
Thickness [mm]	1.42	1.37
Tensile strength [N/mm^2^]	17.80	17.22
Elongation [%]	44	45
Tear strength [N]	90.00	93.83
Brusting factor (lastometer)	8.5	8.5
Adhesion of finish [N/cm]	1.9	1.8
Chromium(III) content converted to Cr_2_O_3_ [%].	3.99	4.06
Sample 1—Hide tanned using a commercial tanning agent		
Sample 2—Hide tanned using chromium recovered from waste subject to acid hydrolysis		

## Data Availability

Not applicable.

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
