# Peer review of "Application of an Integrated System of Thermal Pressure Hydrolysis/Membrane Techniques to Recover Chromium from Tannery Waste for Reuse in Hide Tanning Processes"

_membranes, 2022, doi:10.3390/membranes13010018_

Round 1

Reviewer 1 Report

The manuscript titled "Application of an integrated system of thermal pressure hydrolysis/membrane processes to recover chromium from waste for future reuse in tanneries" describes a new method for chromium recovery from tannery waste. The manuscript is well arranged and in line with the membranes journal. The reviewer's comments are as follows:

1.      The title can be more effective.

2.      The reviewer suggests a revision of the manuscript in terms of grammatical and typo errors.

3.      Section 2. Materials and methods lack a description of materials used in the study with the incorrect numbering of the sub-sections.

4.      Please describe the terms, “JX”, “TS40”, and “DL” in Table 1.

5.      Revise the conclusion. Conclude the manuscript by including the results obtained.

Author Response

Thank you very much for the valuable suggestion and insightful comments contained in the review. Answers to the comments are presented at attachment.

Reviewer 2 Report

This manuscript presented an integrated system of thermal pressure hydrolysis/membrane processes to recover chromium from waste for reuse. There are still some issues that should be noticed and addressed before publication.

1. The image of Fig.12 is not clear enough. The author needs to provide a high-resolution one.

2. Configurations and relative size of inset figures and words in some figures, such as Fig.4 and Fig.13, should be improved. 

Author Response

(The authors gave the same response as above.)
